# Association of Genetic Polymorphisms in SLC45A2, TYR, HERC2, and SLC24A in African Women with Melasma: A Pilot Study

**DOI:** 10.3390/ijms26031158

**Published:** 2025-01-29

**Authors:** Nomakhosi Mpofana, Zinhle Pretty Mlambo, Mokgadi Ursula Makgobole, Ncoza Cordelia Dlova, Thajasvarie Naicker

**Affiliations:** 1Dermatology Department, Nelson R. Mandela School of Medicine, University of KwaZulu-Natal, Durban 4000, South Africa; mokgadim@dut.ac.za (M.U.M.); dlovan@ukzn.ac.za (N.C.D.); 2Department of Somatology, Durban University of Technology, Durban 4000, South Africa; 3Discipline of Optics and Imaging, Doris Duke Medical Research Institute, College of Health Sciences, University of KwaZulu-Natal, Durban 4000, South Africa; zinhlemlambo@gmail.com (Z.P.M.); naickera@ukzn.ac.za (T.N.)

**Keywords:** melasma, hypermelanosis, genetic predisposition, single nucleotide polymorphisms (SNPs)

## Abstract

Melasma is a chronic skin disorder characterized by hyperpigmentation, predominantly affecting women with darker skin types, including those of African descent. This study investigates the association between genetic variants in *SLC45A2*, *TYR*, *HERC2*, and *SLC24A5* genes and the severity of melasma in women of reproductive age. Forty participants were divided into two groups: twenty with facial melasma and twenty without. Deoxyribonucleic acid (DNA) was extracted from blood samples and genotyped using TaqMan assays to identify allele frequencies and genotype distributions. Significant associations were observed for the *TYR* gene (rs1042602), *HERC2* gene (rs1129038), and *SLC24A5* gene (rs1426654) polymorphisms, highlighting their potential roles in melasma susceptibility. For example, the rs1042602 Single Nucleotide Polymorphisms (SNP) in the *TYR* gene showed a strong association with melasma, with the AA genotype conferring a markedly increased risk. Similarly, the rs1129038 SNP in the *HERC2* gene and the rs1426654 SNP in the *SLC24A5* gene revealed significant genetic variations between groups in women of African descent. These findings underscore the influence of genetic polymorphisms on melasma’s pathogenesis, emphasizing the need for personalized approaches to its treatment, particularly for women with darker skin types.

## 1. Introduction

Melasma is an acquired, chronic hypermelanosis facial disorder that affects women with darker skin types in their reproductive age. Women of African, Asian, and Latin American descent classified under Fitzpatrick skin types IV to VI are predominantly affected [1,2]. It is characterized by hyperpigmented macules and patches occurring on sun-exposed areas of the face, such as the cheeks, forehead, and upper lip [1,2]. Globally, the prevalence of melasma ranges from 1% in the general population to 9–50% in high-risk populations [3]. Of note, melasma has a negative impact on an individual’s psychological and emotional well-being because of its visible, disfiguring presentation, which often reduces the individual’s quality of life [2].

The etiology of melasma is not entirely understood [2,4]; however, several factors have been implicated in its pathogenesis that exacerbates the condition. These include ultraviolet (UV) radiation exposure, use of hormonal contraception, hormone replacement therapy, application of certain cosmetics, intake of photosensitizing medications, pregnancy, and psychological stress [4,5]. Additionally, genetic predisposition, thyroid disorders, and certain systemic diseases exacerbate the condition [4,6]. The modified Melasma Area and Severity Index (mMASI) assesses melasma severity. The mMASI score is the sum of the darkness score (D); area score (A); and separate fixed coefficients for the forehead, right malar, left malar, and chin regions (Box 1) [7].

Box 1The mMASI score [6].
Total mMASI score = 0.3 A(f)D(f) + 0.3 A(lm)D(lm) + 0.3 A(rm) D(rm) + 0.1 A(c) D(c)The
range of the total score is from 0 to 24. Area and darkness are scored as
follows:

Area
of involvement:0 =
absent1 =
<10%2 =
10–29%3 =
30–49%4 =
50–69%5 =
70–89%,6
= 90–100%

Darkness:0
= absent1 =
slight,2 =
mild,3 =
marked4 =
severe


### 1.1. Clinical Features

Clinically diagnosed melasma patients have symmetrical, ill-defined hyperpigmented macules on the photo-exposed areas, especially the face (Figure 1) but, occasionally, occur on the upper chest and extremities [8,9]. According to the distribution of these macules, melasma has been classified into three clinical patterns. The centrofacial pattern affects the central face, forehead, nose, cheeks, upper lip, and chin. The involvement of the cheeks and nose characterizes the malar pattern. The mandibular pattern predominantly involves the mandibular dermatome. The most common clinical pattern is the centrofacial type, followed by maxillary melasma and mandibular melasma [8]. True mandibular melasma, i.e., restricted to the ramus of the mandible only, is rare and associated with older individuals and is often related to severe sun exposure [7,8].

Extrafacial melasma is a new, less typical pattern [3]. It occurs on non-facial body parts, including the neck, sternum, and forearms. Melasma affecting the upper limbs has been observed mainly among postmenopausal women, especially women on hormone replacement therapy. This type of melasma resembles facial melasma clinically and histopathologically [11]. Wood’s lamp examination helps differentiate between epidermal and dermal types [12]. The pigmentation is accentuated in the epidermal type and not increased in the dermal type [13]. Dermoscopy has also proven to be beneficial in such melasma categorization where a regular brownish appearance, irregular bluish-gray, and a combination of both are observed in epidermal, dermal, and mixed types, respectively [3].

### 1.2. Key Molecular Pathways in Melasma

Recent studies have highlighted the role of genetic variations, mainly single nucleotide polymorphisms (SNPs), in the melasma pigmentation pathway, suggesting a potential link between specific genotypes and alleles and the severity of melasma [1]. At the molecular level, several pathways are involved in the pathogenesis of melasma, such as the melanocyte-stimulating hormone (MSH)/cyclic adenosine monophosphate (cAMP), the KIT pathway, and the Wnt/β-catenin signaling pathways [14,15]. These pathways play a role in the upregulating of tyrosinase and microphthalmia-associated transcription factors (Figure 2). This upregulation leads to the stimulation of melanogenesis and contributes to the development of melasma [16]. In women of reproductive age, ultraviolet (UV) radiation and hormonal changes activate the microphthalmia-associated transcription factor (MITF) leading to increased melanin synthesis [4,9].

Hormonal fluctuations in estrogen and progesterone during pregnancy or with contraceptive use significantly influence melasma development. Of note, estrogen stimulates melanocyte activity, leading to increased melanin production via pathways involving the melanocortin type 1 receptor (MC1R) and microphthalmia-associated transcription factor (MITF) [4]. Ultraviolet radiation triggers oxidative stress, activating the MITF signaling pathway, which regulates genes involved in pigmentation such as tyrosinase (TYR) and tyrosinase-related proteins such as tyrosinase-related protein 1 (TYRP1) and melan-A (MLANA) [4,18]. The MITF also plays a role in cellular redox balance by regulating genes that manage oxidative stress, thereby impacting melanocyte function and survival [18]. The mitogen-activated protein kinase (MAPK) pathway is intricately linked with MITF phosphorylation, which can lead to its degradation or stabilization depending on the context of the signaling [19]. This pathway also regulates various downstream targets involved in pigmentation and cell proliferation.

Some components of the Wnt/β-catenin pathway have been implicated in melasma as they affect MITF expression and activity, influencing melanocyte differentiation and melanin production [4]. Inflammatory cytokines such as Tumor Necrosis Factor-alpha (TNF-α) modulate MITF expression and activity, contributing to the pathogenesis of melasma. These cytokines can either promote or inhibit melanocyte function based on their concentration and context [19,20]. The phosphoinositide 3-Kinase (PI3K) pathway plays a role in cell survival and growth. It may interact with MITF to promote melanocyte proliferation and survival under stress conditions, further contributing to melasma [19].

The Wingless-related integration site beta-catenin (Wnt/β-catenin) signaling pathway is also integral to melanogenesis. Activation of this pathway promotes melanin production, while its inhibition has been shown to reduce melanocyte activity and melanin synthesis [5]. Additionally, the mitogen-activated protein kinase (MAPK) and phosphatidylinositol-3-kinase/Akt (PI3K/Akt) pathways are implicated in melanin regulation. Agents targeting these pathways, including tyrosinase inhibitors such as arbutin and kojic acid, and depigmenting agents like azelaic acid, have shown efficacy in managing melasma [4,14,15] Personalizing treatment based on the patient’s genetic profile may enhance therapeutic outcomes.

The *SLC45A2*, *TYR*, *HERC2*, and *SLC24A5* genes all play important roles in melanin production, which is essential for pigmentation and melasma development; pathways influenced by these genes are depicted in Table 1. *SLC45A2* encodes a transporter protein that regulates melanosome pH and influences melanin synthesis, and its SNPs (rs11568737, rs16891982, rs28777, and rs183671) have been linked to pigmentation differences [21]. It is also involved in melanosome maturation, and the rs1426654 variant is strongly associated with variations in skin pigmentation. Since *TYR* encodes tyrosinase, the enzyme responsible for the rate-limiting steps in melanin production, and the SNPs rs1042602, rs1393350, and rs1126809 can affect its activity, resulting in altered pigmentation [18]. *HERC2* regulates *OCA2*, another pigmentation gene, and the rs1129038 polymorphism alters OCA2 expression, which influences skin colour [22]. Together, these polymorphisms contribute to melasma by altering melanin production and distribution.

SNPs are the most common form of genetic variation among individuals and can be found throughout the genome [23,24]. SNPs are an essential tool in genetics research, providing valuable information for understanding human genetic variation, disease risk, individual traits, and population dynamics [25]. However, there remains a gap in understanding how these polymorphisms correlate with melasma severity, particularly among populations of African ancestry. Given the unique genetic landscape of the South African (eThekwini) female population, which predominantly consists of individuals with darker skin types, this pilot study explored the correlation between pigment gene polymorphisms of genes *SLC45A2* (rs11568737 and rs28777), *TYR* (rs1042602 and rs1126809), *HERC2* (rs1129038), and *SLC24A* (rs1426654) with the severity of melasma development.

Thus, this study presents a novel investigation into the association of genetic polymorphisms in *SLC45A2*, *TYR*, *HERC2*, and *SLC24A5* in African women with melasma, a population that has been under-represented in genetic research on pigmentation disorders. While most studies on melasma have focused on populations of Asian and Hispanic descent, this research uniquely addresses the possible genetic predispositions in women with darker skin types, contributing to a deeper understanding of the development of melasma in African populations. By exploring these specific polymorphisms, which are critical regulators of melanin biosynthesis and skin pigmentation, the study bridges a significant knowledge gap and offers insights into potential personalized treatment approaches for melasma in this demographic.

## 2. Results

Forty participants were enrolled in the study and assigned to two groups: Group A (women with melasma, *n* = 20) and Group B, women without melasma (control group, *n* = 20). The age of participants ranged between 35 and 58 years (46.80 ± 7.52) in the control group and between 38 and 60 years (47.25 ± 7.99) in the melasma group. Among the women with melasma, 16 (84.21%) had a history of successful pregnancy while 3 (15.79%) did not. In Group A, 2 (10%) had malar melasma, 9 (45%) had centrofacial melasma and 9 (45%) had mandibular melasma. In this group, only 12 (60%) had a familial history (first-degree relative) with melasma while 8 (40%) did not. The duration of melasma affliction was less than 10 years. Clinical characteristics and genotype frequency distribution of SNPs of the study population are shown in Table 2.

### Melasma vs. Control Groups

Genetic variation of the genes *SLC45A2* (rs11568737 and rs28777), *TYR* (rs1042602 and rs1126809), *HERC2* (rs1426654), and *SLC24A* (rs1426654) were assessed. Allelic and genotypic frequency comparisons were conducted, and four genetic models were tested for their association with melasma (Table 2): codominant (equal effect of both alleles in a gene pair), dominant (alleles express the same phenotype), recessive (phenotype is expressed only when both alleles are identical) and overdominant (where the heterozygote has a greater effect than the homozygote). Each of these models was also evaluated for associations with melasma across the six variants studied (Table 2).

The *rs11568737* (G > T) genotype frequencies GG-1 (5%), GT 8-(40%), and TT-11 (55%) were observed in melasma, and GG 2 (10%), GT 2 (10%), and TT 16 (80%) were observed in the control group (Table 2). The allele frequencies G-10 (25%) and T-30 (75%) were observed in the melasma versus G-6 (15%) and T-36 (90%) in the control group (Table 3). The genotypic frequency association of gene polymorphism codominant GG vs. TT showed no significant association in the melasma and the control groups [OR = 0.7273; 95% Cl (0.0464–6.922); adjusted *p* > 0.9999]; GG vs. GT [OR = 0.1250; 95% Cl (0.007879–1.793); adjusted *p* = 0.2028] and GT vs. TT [OR = 5.818; 95% Cl (1.197–29.87); adjusted *p* = 0.2028] (Table 3). Additionally, the allelic frequency association G vs. T was, different between the melasma and the control group (adjusted *p* = 0.4066) (Table 3).

The *rs28777* (A > C) genotype frequencies of AA-2 (10%), AC-15 (75%), and CC-3 (15%) were observed in the melasma group in contrast to AA-6 (30%), AC-12 (60%) and CC-2 (10%) genotype frequencies in the control group. The allele frequencies of A-19 (47.5%) and C-21 (52.5%) were noted in the melasma group compared to A-24 (60%) and C-16 (40%) in the control group (Table 2). Recessive alleles (AA + AC vs. CC: adjusted *p* = 0.0033) were significantly different between the two groups. The allelic frequency association of A vs. C was non-significantly different between the melasma and control groups [OR = 0.6667;95% Cl (0.2834–1.697); adjusted *p* = 0.4949] (Table 2).

The *rs1042602* genotype frequencies of AA-3 (15%), AC-16 (80%), and CC-1 (5%) were observed in the control group compared to AA-2 (10%), AC-4 (20%), and CC-16 (70%) in the melasma group (Table 2). The allele frequencies of A-22 (55%) and C-18 (45%) in the melasma group were different compared to A-8 (20%) and C-32 (80%) in the control group. The genotypic frequency association of gene polymorphism codominant AA vs. CC [OR = 21.00; 95% Cl (1.799–284.1); adjusted *p* = 0.0320] and AC vs. CC [OR = 56.00; 95%Cl (6.496–618.4); adjusted *p* < 0.0001] was significantly associated in both the melasma and control groups. However, the genotypic frequency of AA vs. AC [OR = 0.3750; 95% Cl (0.06110–2.784); adjusted *p* = 0.5623] showed no significant difference between the two groups. Recessive alleles (AA + AC vs. CC; adjusted *p* < 00001) and overdominant alleles (AA + CC vs. AC: adjusted *p* = 0.0449) showed a significant difference between the two groups. The allelic frequency association A vs. C showed a significant difference between the melasma and control groups [OR = 4.889; 95% Cl (1.882–13.78); adjusted *p* = 0.0024] (Table 3).

The *rs1126806* genotype frequencies of GG 15 (75%), GA 2 (10%), and AA 3 (15%) were observed in the melasma group, and GG 14 (70%), GA 4 (20%), and AA 2 (10%) in the control group (Table 2). The allele frequencies G 33 (82.5%) and A 17 (17.5%) were observed in the melasma group and G 30 (75%) and A 10 (25%) in the control group. The genotypic frequency association of gene polymorphism codominant GG vs. AA [OR = 2.143; 95% Cl (0.4153–2.37); adjusted *p* = 0.6581]; GG vs. GA [OR = 0.3571; 95% Cl (0.02590–2.718); adjusted *p* = 0.6074]; and GA vs. AA [OR = 0.7500; 95% Cl (0.03569–11.03); adjusted *p* > 0.9999] showed no significant difference between the two groups. Allelic frequency association G vs. A showed no significant difference between the melasma and control groups [OR = 1.473; 95% Cl (0.5158–4.004); adjusted *p* = 0.5892] (Table 3).

The *rs1129038* genotype frequencies of CC-11 (55%), CT-2 (10%), and TT-7 (35%) were observed in the melasma group compared to CC-2 (10%), CT-17 (85%) and TT-1 (5%) in the control group. The genotypic frequency association of gene polymorphism codominant CC vs. TT [OR = 0.6111; 95% Cl (0.03839–6.044); adjusted; *p* > 0.9999^],^ and CT vs. TT [OR = 0.2857; 95% Cl (0.01344–7.970); adjusted *p* = 0.4909] showed no significant association in the melasma and control groups. However, the genotypic frequency of CC vs. CT [OR = 46.75; 95% Cl (5.786–270.8); adjusted *p* < 0.0001] was significantly different between the two groups. Dominant alleles (CC vs. CT + TT; adjusted *p* = 0.0022), recessive alleles (CC + CT vs. TT: adjusted *p* = 0.0436), and overdominant alleles (CC + TT vs. TC: adjusted *p* < 0.0001) was significantly different between the two groups. Allelic frequency association C vs. T showed no significant difference between the two groups [OR = 1.357; 95% Cl (0.5434–3.117); adjusted *p* = 0.6525] (Table 3).

The *rs1426654* (A > G) genotype frequencies of AA-2 (10%), AG-10 (50%) and GG-8 (40) were observed in the melasma group compared to AA-16 (80%), AG-4 (20%) and GG-0 (0%) in the control group. The allele frequencies A-14 (35%) and G-26 (65%) in the melasma group were different from A-32 (80%) and G-8 (20%) in the control group (Table 2). The genotypic frequency association of gene polymorphism codominant AA vs. GG [OR = 0.03571; 95% Cl (0.005866–0.3303); adjusted *p* = 0.0010] and AA vs. AG [OR = 0.05714; 95% Cl (0.01078–0.3499); adjusted *p* = 0.0022] showed a significant association in the melasma and control groups (Table 3). However, the genotypic frequency of AG vs. GG [OR = 0.6250; 95% Cl (0.1010–3.827); adjusted *p* > 0.9999] showed no significant difference between the two groups. Recessive alleles (AA + AG vs. GG; adjusted *p* = 0.0002) showed a significant difference between the two groups. Allelic frequency association A vs. G showed no significant difference between the two groups (adjusted *p* < 0.0001) (Table 3).

## 3. Discussion

This novel study reports an absence of single nucleotide polymorphisms of the different genotypes (GG vs. TT, GG vs. GT, and GT vs. TT) for rs11568737 of the *SLC45A2* gene in the melasma compared to the control group. The *SLC45A2* gene is involved in melanin production and, hence, affects pigmentation and is highly expressed in melanoma cell lines [26,27]. It is located at 5p13.2 and is composed of seven exons that encode the membrane-associated transporter protein (MATP) [27,28,29]. To the best of our knowledge, this is the first study to demonstrate an absence of association of rs11568737 with melasma in women of African ancestry. Of note, the weak trend may reflect our small study population. Previous reports of genetic variants of rs11568737 have been noted in albinism [27]. Nonetheless, this lack of significant findings in the dominant (GG vs. GT + TT) and recessive (GG + GT vs. TT) alleles supports the absence of any genetic association between the rs11568737 polymorphism of the *SLC45A2* gene with melasma in women of African ancestry.

Our results are in contrast with previous studies which have reported that the *SLC45A2* protein is expressed in melanocyte cell lines and plays a role in melanin synthesis by facilitating tyrosinase trafficking and proton transport to melanosomes [30,31]. Moreover, it controls pH and ionic homeostasis within melanosomes [27,28]. Wang [11] reported that mutations in the *SLC45A2* gene may lead to oculocutaneous albinism type IV (OCA4), while polymorphisms may be linked to darker pigmentation of the skin, hair, and eyes. Also, Abe et al. (2013) reported that SNPs of rs11568737 of the *SLC45A2* gene were significantly associated with melanin index in the Japanese female population [32]. Thus, there remains a paucity of data on the African population, necessitating elucidation of the complex interplay of genetic and environmental factors in these women with melasma.

We also report no significant difference in the codominant, dominant, and overdominant allele models for rs28777 of the *SLC45A2* gene. However, we report a significant association (*p* = 0.0033) in the recessive allele model (AA + AC vs. CC), suggesting that the CC genotype may be a risk factor for melasma development in women of African descent. Single nucleotide polymorphism of rs28777 (A > C) of the *SLC45A2* gene has been reported to influence skin pigmentation [15]. Alterations in this gene lead to changes in the activity of the *SLC45A2* protein, affecting melanin production [16]. Nonetheless, it must be noted that studies on the rs28777 polymorphism in melasma are limited while the complex interactions between multiple genes and environmental factors in melasma suggest a possible link.

We also report a significant association between the melasma vs. control groups in the codominant allele models for rs1042602 of the *TYR* gene, particularly in the CC genotype. However, the absence of a significant association in the other allelic genotypic models indicates that the rs28777 polymorphism alone may not be a strong independent predictor of melasma risk. We also report a significant association in the codominant allele models for rs1042602 of the *TYR* gene, particularly in the CC genotype, between the melasma vs. control groups. Notably, the comparison of AA vs. CC genotypes (OR = 21.00; 95% CI: 1.799–284.1; *p* = 0.0320) and AC vs. CC (OR = 56.00; 95% CI: 6.496–618.4; *p* < 0.0001) demonstrate strong associations, indicating that the CC genotype significantly increases the risk of developing melasma. The significant association in the recessive allele model (AA + AC vs. CC, *p* < 0.0001) further supports the fact that the CC genotype affects melasma susceptibility. Additionally, the overdominant allele model (AA + CC vs. AC) also showed a significant difference (*p* = 0.0449), suggesting that the AC genotype may have a protective effect against melasma development. The allelic frequency comparison (A vs. C) highlights the fact that the C allele was more prevalent in the melasma group (OR = 4.889; 95% CI: 1.882–13.78; *p* = 0.0024). The rs1042602 is found in the *TYR* gene, which encodes the tyrosinase enzyme and is crucial for melanin production and linked to skin pigmentation [33,34]. The absence of tyrosinase exerts an epistatic effect on downstream pigment-related genes, effectively stopping melanin production [35,36].

The significant association of the C allele with melasma observed in our study suggests a potential genetic marker for identifying susceptible women of African descent while also demonstrating a pathogenic link between the rs1042602 genetic variant and skin pigmentation. The SNP rs1042602 is a C to A transversion within the coding region of the *TYR* gene leading to an S192Y mutation, which is associated with eye, hair, and skin pigmentation in several populations [37,38]. It has been reported to be associated with pigmentation in general and with eye and hair pigmentation in European populations [35]. Similarly, the rs1042602 A/192Tyr allele was shown to be strongly associated with lighter skin color eye color, and absence of freckles in a population of South Asian ancestry in Europe, [35,39,40]. The rs1042602 in *TYR* has also been reported to be significantly associated with skin color in Brazilian populations [37].

Findings from this study did not reveal any significant differences between the two study groups, suggesting that the rs1126809 of the *TYR* gene does not play a major role in melasma susceptibility. The analysis of the genotypic associations also showed no significant differences in any of the allele models tested. For instance, the codominant model comparisons of GG vs. AA genotypes (OR = 2.143; 95% CI: 0.4153–12.37; *p* = 0.6581), GG vs. GA (OR = 0.3571; 95% CI: 0.02590–2.718; *p* = 0.6074), and GA vs. AA genotypes (OR = 0.7500; 95% CI: 0.03569–11.03; *p* > 0.9999) all indicated no significant associations. Similarly, the dominant, recessive, and overdominant allele frequency models showed no significant associations between melasma and the control group. The rs1126809 (G > A) polymorphism is located in the *TYR* gene, which plays a critical role in melanin production [41]. The G > A transition results in an amino acid substitution from Arg402Gln in the tyrosinase protein [41,42]. Arg402Gln has been described to be associated with eye color and skin type [41].

The significant associations observed in our study suggest that SNPs of rs1129038 of the *HERC2* gene may influence the expression of other pigment-related genes, thereby contributing to the risk of melasma development. In the melasma group, the genotype frequencies demonstrate a significant association between the CC and CT genotypes (OR = 46.75; 95% CI: 5.786–270.8; *p* < 0.0001), indicating that the CC genotype is more prevalent in the melasma group compared to the CT genotype of the control group. Furthermore, the dominant allele model (CC vs. CT + TT) also showed a significant association with melasma development (*p* = 0.0022). The findings were similar for the recessive (CC + CT vs. TT, *p* = 0.0436) and overdominant (CC + TT vs. CT, *p* < 0.0001) allele models. The rs1129038 polymorphism is located near the *HERC2* gene, which regulates the expression of the OCA2 gene, a critical role player in melanin production [43,44]. Notably, variants in this region have been implicated in various pigmentation traits, including eye color and susceptibility to hyperpigmentation disorders [44]. Our findings suggest that the CC genotype may increase the risk of melasma development, while the CT genotype may have a protective effect. However, the allelic frequency association between the C and T alleles showed no significant difference between the melasma and control groups (OR = 1.357; 95% CI: 0.5434–3.117; *p* = 0.6525), indicating that the overall allele distribution may not be a strong predictor of melasma susceptibility.

Our findings show a significant association between the AA genotype and the GG genotype for rs1426654 of the SLC24A5 gene with melasma (OR = 0.03571; 95% CI: 0.005866–0.3303; *p* = 0.0010), suggesting that the GG genotype may confer a higher risk for melasma development in women of African ancestry. Similarly, the AA vs. AG genotype comparison also demonstrated a significant association (OR = 0.05714; 95% CI: 0.01078–0.3499; *p* = 0.0022). However, the AG vs. GG genotype comparison did not show a significant difference (OR = 0.6250; 95% CI: 0.1010–3.827; *p* > 0.9999), indicating that the AG genotype may not significantly influence melasma risk when compared to the GG genotype. The recessive allele model (AA + AG vs. GG) did show a significant association (*p* = 0.0002), further supporting the potential role of the GG genotype in increasing susceptibility to melasma. However, the dominant allele model (AA vs. AG + GG; *p* = 0.0648) and the overdominant allele model (AA + GG vs. AG; *p* = 0.0958) were non-significantly associated. The analysis of allelic frequency showed a significant difference between the melasma and control groups, with the G allele being more prevalent in the melasma group (*p* < 0.0001).

The rs1426654 polymorphism of the *SLC24A5* gene has been extensively studied in skin pigmentation [45]. The SNP rs1426654 in the *SLC24A5* gene involves a guanine-to-adenine substitution in exon three (G>A), leading to an amino acid change from alanine (Ala) to threonine (Thr) at position 111 of the protein (Ala111Thr) [45,46]. This alteration results in a protein with diminished ion transport efficiency, which, in turn, affects pheomelanin production [47,48]. Previous research identified the rs1426654 variant as a major determinant of skin pigmentation differences. This rs1426654 gene variant has been reported to be found almost ubiquitously in Western European populations but rarely in dark-skinned non-European populations [45]. Additionally, it has been reported to have a significant association with skin pigmentation within those of West Maharashtra (Indian) descent where it is thought to play a major role in shaping pigmentation variation [49].

According to previously reported studies, the *SLC24A5* gene plays a significant role in pigmentation [50,51]. In African and East Asian populations, the ancestral genotype G of rs1426654 is highly conserved, with frequencies ranging from 93 to 100%, contributing to darker skin [51]. In contrast, the variant rs1426654/A is predominant in Europeans, with frequencies ranging from 98 to 100%, leading to fairer phenotypes [47,48,51]. Our results correlate with findings that report the A allele frequency (associated with light skin) is low (0.444) among the Warl tribe of India compared to those of darker skin pigmentation [49]. While the exact mechanism by which the rs1426654 variant contributes to melasma development remains unclear, this polymorphism may alter melanin synthesis, leading to an increased susceptibility to hyperpigmentation under certain environmental conditions, such as high sun exposure. It has been previously reported that the homozygous AA genotype (rs1426654/A) is strongly associated with fairer skin phenotypes, with this association being particularly pronounced in individuals with white skin (OR 47.8; CI 14.1–161.6; *p* < 0.0001) compared to those who self-identify as black-skinned [48]. The heterozygous genotype GA of rs1426654 also exhibits a significant association with European Americans compared to African Americans with frequencies of 8.6% and 1.4%, respectively [47,48]. This difference suggests a potential racial disparity in the genetic predisposition to certain conditions, for example, melasma.

The findings of this study provide a critical contribution to the limited body of research on the genetic basis of melasma in African women. By examining polymorphisms in *SLC45A2*, *TYR*, *HERC2*, and *SLC24A5*, this pilot study highlights the potential genetic factors that may influence melasma susceptibility and severity in a population with distinct pigmentation characteristics. These results underscore the importance of including diverse populations in genetic studies and pave the way for future research to explore targeted interventions and treatments tailored to the unique genetic and phenotypic profiles of African women affected by melasma.

The significant association of rs1042602 and rs1126809 polymorphisms with melasma suggests that these genetic variants may play a crucial role in the susceptibility to the skin disorder, particularly in individuals with specific genotypes. These findings highlight the potential for genetic screening to identify individuals at higher risk, allowing for tailored treatment approaches targeting the underlying genetic mechanisms, such as tyrosinase (TYR) activity and melanin production. Moreover, the study underscores the importance of understanding genetic predisposition in diverse populations, especially those with skin of color where melasma is more prevalent. This knowledge could enhance clinical management of the melasma by guiding more effective and personalized treatment options while also paving the way for future research into gene–environment interactions and the development of novel therapies. We acknowledge that while these findings are preliminary, they offer valuable starting points for further investigation into genetic markers that could inform clinical practice

### Limitations and Significance of the Study

The small sample size of this study limits the statistical power and generalizability of our findings. The study was carried out to generate preliminary data, and future studies with larger sample sizes are needed to confirm the associations observed. Since this is a pilot study, our findings require validation in larger, more diverse populations. Additionally, the study may not fully account for environmental and dietary factors influencing melasma, such as the consumption of tyrosine- and tryptophan-rich foods, which may influence melanin production. Future research should consider incorporating nutritional assessments to better understand the potential impact of diet on melasma and related genetic studies. However, despite these limitations, this pilot novel study demonstrates valuable insights into melasma development in women of African descent by exploring the genetic role of specific SNPs (*SLC45A2*, *TYR*, *HERC2*, and *SLC24A5*) in this under-represented population. The interdisciplinary approach of our study and focus on pigmentation-related genes could result in the development of personalized treatment for African women plagued with melasma.

## 4. Materials and Methods

### 4.1. Ethics Statement

The study was carried out according to the 1964 Declaration of Helsinki recommendations and its respective amendments and recommendations on 2015 ICH E6 (R1) Good Clinical Practice. This study received institutional ethics approval (BREC/00002721/2021). All study participants provided written informed consent from four private dermatology clinics in eThekwini, KwaZulu-Natal, South Africa, during the period of March–December 2023.

### 4.2. Study Participants and Clinical Examination

The study population consisted of 40 unrelated women of African descent. Participants were required to confirm their ethnic background, ensuring they were of African descent. The non-relatedness was verified by asking participants for their clan names as clan names often correspond to specific familial or regional lineages. If the clan names differed, the participants were considered non-related and eligible for the study. However, if the clan names matched, it indicated that they were related, and they were excluded from participation. The groups were divided into those with facial melasma (Group A, *n* = 20) and those without facial melasma (Group B, *n* = 20). Controls (Group B) were paired with cases according to age group (±5 years). Melasma was classified from mild to severe using the modified Melasma Area Severity Index (mMASI) score. All participants in both groups used contraceptives or hormone therapy.

### 4.3. DNA Extraction and Genotyping

Whole blood samples were obtained using the EDTA-containing vacutainer tubes (*n* = 40). DNA was extracted from 200 µL of whole blood samples using the QIA^®^amp DNA Blood Mini Kit as per the manufacturer’s instructions (QIAGEN, Valencia, CA, USA). Following extraction, DNA was eluded with nuclease-free water. Then, we used the nanodrop to check the concentration and the purity of DNA. We used spectrophotometry (A260/A280 ratio), which ranged from 1.7 to 2, which indicated good-quality DNA. These steps ensured the reliability of the extracted DNA for subsequent genotyping and the DNA samples were stored at −20 °C until genotyping analysis was performed.

Using a TaqMan pre-designed SNP genotyping assay, six SNPs Reference SNP cluster ID rs11568737 (C__58344129_10), rs28777 (C__2390574_10), rs1042602 (C__8362862_10), rs1126809 (C__17567502_10), rs1129038 (C__489033_10) and rs1426654 (C__2908190_10 were genotyped according to the manufacturer’s protocol (Applied Biosystems by ThermoFisher Scientific, Foster City, CA, USA). Purified DNA samples were amplified using a TaqMan master mix (Applied Biosystems by ThermoFisher Scientific) following the manufacturer’s protocol. The genotyping was performed on a QuantStudio 7 Flex Real-Time PCR System (Applied Biosystems by Thermofisher Scientific). The reaction master mix comprised 0.25 µL of a 20X working stock of TaqMan SNP genotyping assay, 2.5 µL of 2X TaqMan Universal Master Mix, and 3 µL of DNA, resulting in a total volume of 5.75 µL per well.

The TaqMan genotyping assay used two fluorescently labeled primers to discriminate between the two alleles of each SNP. One primer was labeled with VIC^®^ dye (a green fluorophore) for the wild-type allele, while the other was labeled with 6-carboxyfluorescein (6-FAM™) dye (a blue fluorophore) for the mutant allele. Additionally, the primers contained a minor groove binder and a non-fluorescent quencher on the 3′ ends. Following PCR amplification, genotype and allelic discrimination results were analyzed using QuantStudio™ Design and Analysis Software version 1.5.2. An Excel database was subsequently created to compare the SNP genotypes in Group A versus Group B.

### 4.4. Statistical Analysis

The statistical analysis was performed using GraphPad Prism 5 (GraphPad Software, San Diego, CA, USA). The Mann–Whitney test was employed to determine the statistical significance of differences in clinical features. The Hardy–Weinberg equilibrium test was applied to assess genotype distribution conformity in both the control and case groups. Additionally, the chi-square test was utilized for subgroup comparisons. The strength of the association was measured using odds ratios (ORs) with 95% confidence intervals (CIs). A *p*-value of less than 0.05 was considered the threshold for statistical significance.

The Hardy–Weinberg equilibrium (HWE) test was utilized to assess the conformity of observed genotype frequencies. The presence of genotypes was described using frequency and percentage metrics. The strength of associations was expressed as odds ratios (OR) along with 95% confidence intervals (CI) for categorical data, while the Wilcoxon rank-sum test was applied for numerical data. A *p*-value of less than 0.05 was deemed statistically significant. Demographic analyses were conducted using a t-test with GraphPad Prism 8.43 software (GraphPad Software, San Diego, CA, USA), and multiple comparisons were adjusted using the Bonferroni correction test.

## 5. Conclusions

The rs1042602 polymorphism of the *TYR* gene was strongly associated with melasma development, particularly the CC genotype, indicating its potential role in developing hyperpigmentation disorders. In contrast, the rs1126809 of the *TYR* gene did not exhibit significant associations with melasma. Polymorphisms rs1129038 of the *HERC2* gene and rs1426654 (A > G) of the *SLC24A5* gene also showed significant associations with melasma, particularly for the CC and GG genotypes, respectively, suggesting their involvement in melasma risk. The current study showed no significant association between rs11568737 polymorphism of the *SLC45A2* gene with melasma development, while the rs28777 (A > C) polymorphism of the same gene revealed a significant link in the recessive allele model, suggesting increased risk for individuals with the CC genotype. These findings from this study underscore the necessity for tailored treatment approaches that consider genetic variations. Further research with larger sample sizes and a focus on gene–gene and gene–environment interactions is warranted to fully elucidate melasma’s genetic underpinnings.

## Figures and Tables

**Figure 1 ijms-26-01158-f001:**
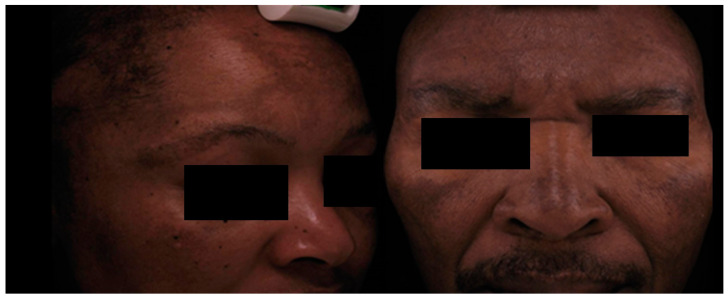
Clinical features of facial melasma. Female full-face involvement (**left**) and male pattern melasma (**right**) [10].

**Figure 2 ijms-26-01158-f002:**
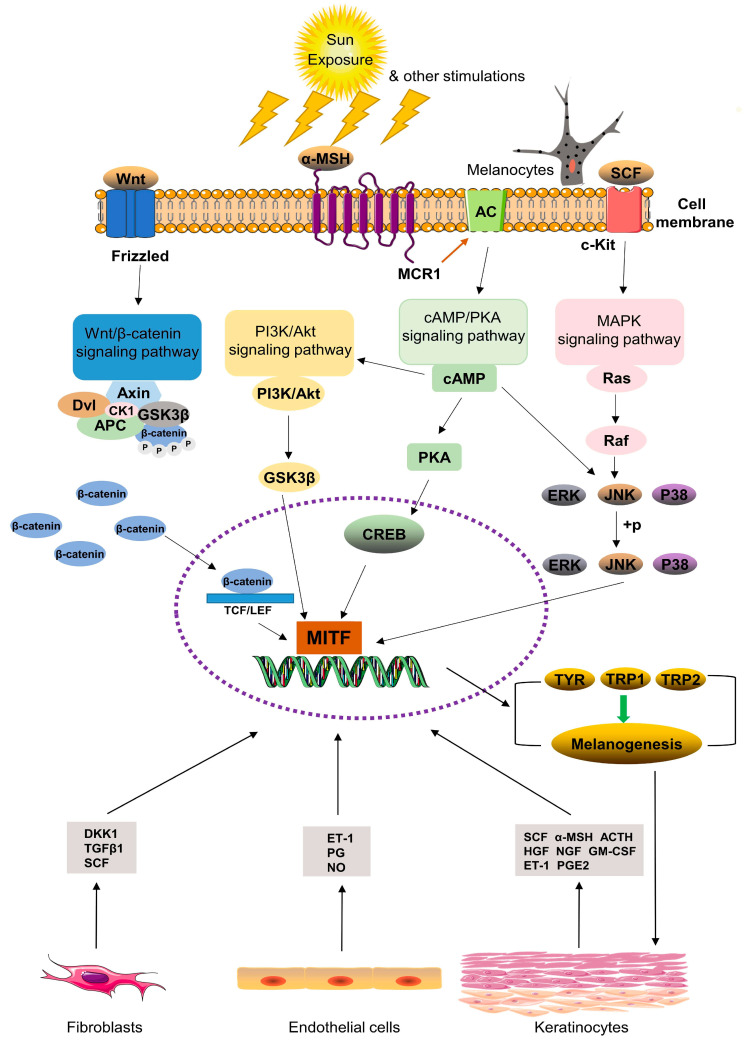
Common signal pathways involved in melanogenesis [17]. Downward arrows represent the transmission or propagation of a signal from an upstream component to a downstream component in the pathway Abbreviations: α-MSH, α-melanocyte-stimulating hormone; MCR1, melanocortin 1 receptor; AC, adenylate cyclase; cAMP, cyclic adenosine monophospha te; PKA, protein kinase A; CREB, cAMP response element-binding protein; PI3K, phosphatidylinositol-3-kinase; GSK3β, glycogen synthase kinase-3β; SCF, stem cell factor; c-Kit, receptor tyrosine kinase; ERK, extracellular signal-regulated kinases; JNK, c-Jun N-terminal kinase; Axin, axis inhibition; Dvl, Dishevelled; CK1, casein kinase 1; APC, adenomatous polyposis coli tumor suppressor protein; TCF/LEF, T cell factor/lymphocyte enhancer factor-1; MITF, microphthalmia-associated transcription factor; TYR, tyrosinase; TRP1, tyrosinase-related proteins-1; TRP2, tyrosinase-related proteins-2; SCF, stem cell factor; ACTH, adrenocorticotropic hormone; HFG, hepatocyte growth factor; NGF, hepatocyte growth factor; GM-CSF, granulocyte-macrophage colony-stimulating factor; ET-1, endothelin 1; PGE2, prostaglandin E2; DKK1, Dickkopf-related protein 1; TGF-β1, transforming growth factor-β1; PG, prostaglandin; NO, nitrogen monoxide [17].

**Table 1 ijms-26-01158-t001:** Genes, pathways, and their role in melanogenesis.

Gene	Pathway Influence	Role in Melanogenesis	Reference
*SLC45A2*	Melanin synthesis and transport	Maintains melanosome pH; essential for tyrosinase activity; mutations lead to reduced pigmentation (OCA4)	[21]
*TYR*	Melanin biosynthesis	Encodes tyrosinase, the key enzyme for melanin biosynthesis; mutations affect pigmentation levels	[18]
*HERC2*	Regulates OCA2 expression	Regulates OCA2 expression; the interaction between *HERC2* and OCA2 influences melanosome function (affects melanosomal pH) function and pigmentation traits (overall melanin production)	[22]
*SLC24A5*	Melanosome maturation	Involved in ion transport for melanosome maturation and sensitivity to UV exposure	[21]

**Table 2 ijms-26-01158-t002:** Participants’ characteristics and genotype/allele frequency distribution of SNPs.

Characteristics	Melasma Group (*n* = 20)	Control Group (*n* = 20)	SNP ID	Genotype	Melasma Group (%)	Control Group (%)	Allele	Melasma Group (%)	Control Group (%)
Age (years)	47.25 ± 7.99	46.80 ± 7.52	***SLC45A2*—rs11568737 (G > T)**	GG	1 (5%)	2 (10%)	G Major	10 (25%)	6 (15%)
History of pregnancy (%)	Yes: 16 (84.21%)	Yes: 12 (60%)	GT	8 (40%)	2 (10%)	T Minor	30 (75%)	36 (90%)	
	No: 3 (15.79%)	No: 8 (40%)	TT	11 (55%)	16 (80%)				
Use of contraceptives/hormone therapy (%)	Yes: 20 (100%)	Yes: 20 (100%)	***SLC45A2*—rs28777 (A > C)**	AA	2 (10%)	6 (30%)	A Major	19 (48%)	24 (60%)
Location of Melasma (%)	Malar: 2 (10%)	-	AC	15 (75%)	12 (60%)	C Minor	21 (53%)	16 (40%)	
	Centrofacial: 9 (45%)	-	CC	3 (15%)	2 (10%)				
	Mandibular: 9 (45%)	-	***TYR*—rs1042602 (A > C)**	AA	3 (15%)	2 (10%)	A Major	22 (55%)	8 (20%)
Duration of Melasma (%)	2 years: 5 (25%)	-	AC	16 (80%)	4 (20%)	C Minor	18 (45%)	32 (80%)	
	5 years: 7 (35%)	-	CC	1 (5%)	14 (70%)				
	10 years: 4 (20%)	-	***TYR*—rs1126809 (G > A)**	GG	15 (75%)	14 (70%)	G Major	33 (83%)	30 (75%)
	More than 10 years: 4 (20%)	-	GA	2 (10%)	4 (20%)	A Minor	7 (18%)	10 (25%)	
Family History (First-degree relative) (%)	Yes: 12 (60%)	-	AA	3 (15%)	2 (10%)				
	No: 8 (40%)	-	***HERC2*—rs1129038 (C > T)**	CC	11 (55%)	2 (10%)	C Major	24 (60%)	21 (53%)
mMASI Score	Mild: 5 (25%)	-	CT	2 (10%)	17 (85%)	T Minor	16 (40%)	19 (48%)	
	Moderate: 7 (35%)	-	TT	7 (35%)	1 (5%)				
	Severe: 8 (40%)	-	***SLC24A5*—rs1426654 (A > G)**	AA	2 (10%)	14 (70%)	A Major	14 (35%)	32 (80%)
			AG	10 (50%)	4 (20%)	G Minor	26 (65%)	8 (20%)	
			GG	8 (40%)	2 (10%)				

**Table 3 ijms-26-01158-t003:** Genotypic and allelic associations of gene polymorphisms.

SNP		Melasma vs. Control OR (95% CI), *p*-Value
** *SLC45A2* ** **-rs11568737 ** **G > T Genotype**		
**Codominant**	AA vs. TT	0.7273 (0.04641–6.922) *p* > 0.9999
	GG vs. GT	0.1250 (0.007879–1.793) *p* = 0.2028
	GT vs. TT	5.818 (1.197–29.87) *p* = 0.0625
**Dominant**	GG vs. GT + TT	0.5263 (0.03471–4.882) *p* > 0.9999
**Recessive**	GG + GT vs. TT	3.682 (0.9184–12.49) *p* = 0.0956
**Overdominant**	AA + CC vs. AC	0.1667 (0.03278–0.8302) *p* = 0.0648
**Allele** **(Major vs. minor)**	G vs. T	1.771 (0.5885–5.690) *p* = 0.4066
** *SLC45A2* ** **-rs28777** **A > C** **Genotype**		
**Codominant**	AA vs. CC	0.2222 (0.02725–1.978) *p* = 0.2929
	AA vs. AC	0.2667 (0.04972–1.485) *p* = 0.2285
	AC vs. CC	0.9444 (0.1501–5.220) *p* > 0.9999
**Dominant**	AA vs. AC + CC	0.2593 (0.04908–1.366) *p* = 0.2351
**Recessive**	AA + AC vs. CC	0.1111 (0.02356–0.4805) *p* = 0.0033 **
**Overdominant**	AA + CC vs. AC	0.4412 (0.1185–1.707) *p* = 0.3200
**Allele** **(Major vs. minor)**	A vs. C	0.6667 (0.2834–1.697) *p* = 0.4949
** *TYR* ** **-rs1042602** **A > C** **Genotype**		
**Codominant**	AA vs. CC	21.00 (1.799–284.1) *p* = 0.0320 *
	AA vs. AC	0.3750 (0.06110–2.784) *p* = 0.5623
	AC vs. CC	56.00 (6.496–618.4) *p* < 0.0001 ****
**Dominant**	AA vs. AC + CC	1.412 (0.2581–8.679) *p* > 0.9999
**Recessive**	AA + AC vs. CC	44.33 (4.824–487.8) *p* < 0.0001 ****
**Overdominant**	AA + CC vs. AC	0.1667 (0.03149–0.9046) *p* = 0.0449 *
**Allele** **(Major vs. minor)**	A vs. C	4.889 (1.882–13.78) *p* = 0.0024 **
** *TYR* ** **-rs1126809** **G > A** **Genotype**		
**Codominant**	GG vs. AA	2.143 (0.4153–12.37) *p* = 0.6581
	GG vs. GA	0.3571 (0.02590–2.718) *p* = 0.6074
	GA vs. AA	0.7500 (0.03569–11.03) *p* > 0.9999
**Dominant**	GG vs. GA + AA	0.6429 (0.1501–3.368) *p* = 0.7013
**Recessive**	GG + GA vs. AA	2.250 (0.4530–12.80) *p* = 0.6614
**Overdominant**	GG + AA vs. GA	0.6296 (0.1032–3.432) *p* > 0.9999
**Allele** **(Major vs. minor)**	G vs. A	1.473 (0.5158–4.004) *p* = 0.5892
** *HERC2* ** **-rs1129038** **C > T** **Genotype**		
**Codominant**	CC vs. TT	0.6111 (0.03839–6.044) *p* > 0.9999
	CC vs. CT	46.75 (5.786–270.8) *p* < 0.0001 ****
	CT vs. TT	0.2857 (0.01344–7.970) *p* = 0.4909
**Dominant**	CC vs. CT + TT	12.38 (2.162–61.41) *p* = 0.0022 **
**Recessive**	CC + CT vs. TT	0.09774 (0.008320–0.7323) *p* = 0.0436 *
**Overdominant**	CC + TT vs. CT	51.00 (6.786–262.2) *p* < 0.0001 ****
**Allele** **(Major vs. minor)**	C vs. T	1.357 (0.5434–3.117) *p* = 0.6525
** *SLC24A5* ** **-rs1426654** **A > G** **Genotype**		
**Codominant**	AA vs. GG	0.03571 (0.005866–0.3303) *p* = 0.0010 **
	AA vs. AG	0.05714 (0.01078–0.3499) *p* = 0.0022 **
	AG vs. GG	0.6250 (0.1010–3.827) *p* > 0.9999
**Dominant**	AA vs. AG + GG	0.1667 (0.03278–0.8302) *p* = 0.0648
**Recessive**	AA + AG vs. GG	0.04762 (0.009685–0.2782) *p* = 0.0002 ***
**Overdominant**	AA + GG vs. AG	0.2500 (0.07356–1.000) *p* = 0.0958
**Allele** **(Major vs. minor)**	A vs. G	0.1346 (0.05049–0.3585) *p* < 0.0001 ****

OR: odds ratio; CI: confidence intervals; Asterisks (*) denote significance: * *p* < 0.05, ** *p* < 0.01, *** *p* < 0.001 and **** *p* < 0.0001

## Data Availability

All data are provided in the manuscript.

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
