# Peer review of "Association of Genetic Polymorphisms in SLC45A2, TYR, HERC2, and SLC24A in African Women with Melasma: A Pilot Study"

_ijms, 2025, doi:10.3390/ijms26031158_

Round 1
Reviewer 1 Report
Comments and Suggestions for Authors
The manuscript " Association of SLC45A2 (rs11568737, rs28777); TYR (rs1042602, rs1126809); HERC2 (rs1129038) and SLC24A (rs1426654) Single Nucleotide Polymorphisms with Melasma in African Women – A Pilot Study" aims to investigate the association between genetic variants in SLC45A2, TYR, HERC2, and SLC24A genes and melasma severity in women with darker skin, emphasizing personalized treatment approaches. Although the paper presents a certain scientific interest, there are some concerns regarding the data's validity and overall results. Here are some important comments:
1. The title seems wordy and should have a shorter version: "Association of Genetic Polymorphisms in SLC45A2, TYR, HERC2, and SLC24A with Melasma in African Women: A Pilot Study."
2. A sample size of 40 participants, 20 in each group, is small and diminishes the generalization of results. Justification of sample size may be presented in the limitation section.
3. Expand on the selection criteria for "unrelated women of African descent." Indicate if there was consideration regarding genetic heterogeneity within this population.
4. As all subjects had contraceptive or hormone therapy, explain how these variables were controlled or analyzed because they may be a factor in the development and severity of melasma.
5. Give more information about how DNA quality and quantity were checked after extraction to give reliable results from genotyping.
6. The results section is not very clear because of too much detail in numeric data. Key findings should be summarized in concise tables or bullet points, focusing on statistically significant outcomes.
7. State the practical or clinical significance of findings that reached significance, such as the association of rs1042602 and rs1126809 polymorphisms with melasma.
8. More comparisons should be made with findings from similar studies in other populations to validate or contrast the current results.
9. The English language needs further improvement.
Comments on the Quality of English Language
The English language needs further improvement.
Author Response
Please find attached the responses to reviewer comments.
Thank you

Reviewer 2 Report
Comments and Suggestions for Authors
I have carefully reviewed the article titled “Association of SLC45A2 (rs11568737, rs28777); TYR 2 (rs1042602, rs1126809); HERC2 (rs1129038) and SLC24A 3 (rs1426654) Single Nucleotide Polymorphisms with 4 Melasma in African Women – A Pilot Study” In which the authors evaluate polymorphisms of several genera to determine whether or not they are risk factors for melasma in African women.
The study is interesting and well-posed to understand the genetic factor that contributes to the development of melasma. However, from my perspective, there are several points that are not clear, are confusing, and that the authors should improve.
1. The sample size is too small, so the statistical power must be low. What is the statistical power of your study? How do you justify the low sample size and, if applicable, the low statistical power?
2. In table 2, the word “contraceptives” is misspelled
3. In Table 3, the "patients" category covers both cases and controls, the patients category for cases and the control category must be separated.
4. In Table 3, although the polymorphisms are listed, it is not indicated to which gene each polymorphism belongs, and that makes the table confusing. I suggest that a small title be included listing each gene, followed by its polymorphisms.
5. In table 3, the result 14 (35%, is missing the sign to close the parentheses
6. In tables 2 and 3 they should be a single table indicating the N, allelic frequencies, genotypic frequencies, OR, 95% CI and p value, since separate tables make the results confusing and difficult to understand. In a single table, the reader will appreciate from the beginning the genetic models that were associated for each polymorphism.
7. The p-values obtained in Tables 2 and 3 are not corrected by the Bonferroni test. This correction is necessary to display the true p-value. Another column should also be placed after the p-value indicating pc (corrected p-value).
8. What Hardy Weinberg equilibrium value did the researchers obtain? They mention it but do not provide the numerical value. It is essential to know this value given the low sample size.
9. The results describe exactly the same thing as what is observed in the tables, I suggest being briefer in the description of results since the work becomes repetitive.
10. What do the authors mean by a pilot study? So a pilot study is less valuable than a normal study? So in a pilot study, the results are unreliable?
11. The researchers do not state the origin of the probes acquired (Thermofisher?). The researchers should not only state the type of probes used, but also the rs and C__ number.
12. The statistical analysis section talks little about statistics. Here the authors should include the programs and statistical tests they used for each statistical analysis and not speak in general terms.
13. Why did the authors use ANOVA for demographic analysis, if they only have 2 groups?
Author Response
Please find attached the response to reviewer comments
Thank you

Reviewer 3 Report
Comments and Suggestions for Authors
Major Comments:
The authors need to address the following queries:
1. Please mention the novelty of the study.
2. Do the authors have any data from male patients?
3. How variations in different ethnic groups and races would influence on the Melasma? Please opine.
4. The ‘Abstract’ is full of statistics. Instead, the authors should mention the principal result/findings of the study.
5. In the ‘Introduction’ vivid description including the figure on the known key molecular pathways in Melasma seems to be redundant!
6. How did the authors select the ‘Control’ subjects? Are they from the same family of the corresponding patients?
7. The sample size of the study is low!
8. Can the food habit (especially tyrosine/ tryptophan rich food) of the patient effect on the outcome of the study?
Minor Comments:
1. The authors should thoroughly check the manuscript for the grammatical errors (e.g. Line 589: ‘Form’??).
Comments on the Quality of English LanguageThe authors should thoroughly check the manuscript for the grammatical errors (e.g. Line 589: ‘Form’??).
Author Response
Please find attached the response to reviewer comments.
Thank you

Round 2
Reviewer 1 Report
Comments and Suggestions for Authors
After thoroughly reviewing the revised manuscript and considering the authors' revisions and responses to the referee's comments, I find that the manuscript has been significantly improved. The authors have effectively addressed the concerns, enhancing their study's clarity and scientific rigor. The revisions have clarified the methodology, improved the presentation of results, and strengthened the discussion and conclusions.
Therefore, I believe that the manuscript now meets the standards required for publication in IJMS and I recommend that it be accepted for publication.
Thank you for considering my recommendation.
Comments on the Quality of English LanguageThe English language has undergone improvements and has gained wider acceptance.
Reviewer 2 Report
Comments and Suggestions for Authors
The researchers have made all the suggested changes, now their manuscript is clearer and more organized. I suggest it be published.